# In Vitro Cytotoxicity of Antiresorptive and Antiangiogenic Compounds on Oral Tissues Contributing to MRONJ: Systematic Review

**DOI:** 10.3390/biom13060973

**Published:** 2023-06-10

**Authors:** Robert H. Guirguis, Leonard P. Tan, Rebecca M. Hicks, Aniqa Hasan, Tina D. Duong, Xia Hu, Jordan Y. S. Hng, Mohammad H. Hadi, Henry C. Owuama, Tamara Matthyssen, Michael McCullough, Federica Canfora, Rita Paolini, Antonio Celentano

**Affiliations:** 1Melbourne Dental School, The University of Melbourne, 720 Swanston Street, Carlton, VIC 3053, Australia; rguirguis@student.unimelb.edu.au (R.H.G.); leonardt1@student.unimelb.edu.au (L.P.T.); rita.paolini@unimelb.edu.au (R.P.); 2Department of Neuroscience, Reproductive Sciences and Dentistry, University of Naples Federico II, 80131 Naples, Italy; federica.canfora@unina.it

**Keywords:** MRONJ, bisphosphonates, antiresorptives, antiangiogenics, oral tissues

## Abstract

Background: Invasive dental treatment in patients exposed to antiresorptive and antiangiogenic drugs can cause medication-related osteonecrosis of the jaw (MRONJ). Currently, the exact pathogenesis of this disease is unclear. Methods: In March 2022, Medline (Ovid), Embase (Ovid), Scopus, and Web of Science were screened to identify eligible in vitro studies investigating the effects of antiresorptive and antiangiogenic compounds on orally derived cells. Results: Fifty-nine articles met the inclusion criteria. Bisphosphonates were used in 57 studies, denosumab in two, and sunitinib and bevacizumab in one. Zoledronate was the most commonly used nitrogen-containing bisphosphonate. The only non-nitrogen-containing bisphosphonate studied was clodronate. The most frequently tested tissues were gingival fibroblasts, oral keratinocytes, and alveolar osteoblasts. These drugs caused a decrease in cell proliferation, viability, and migration. Conclusions: Antiresorptive and antiangiogenic drugs displayed cytotoxic effects in a dose and time-dependent manner. Additional research is required to further elucidate the pathways of MRONJ.

## 1. Introduction

Medication-related osteonecrosis of the jaw (MRONJ) is a rare and severe adverse drug reaction caused by antiresorptive and antiangiogenic drugs [1]. The overall incidence of MRONJ varies widely across different studies, being reported from 0.01% after oral use of low-dose bisphosphonates to 14.4% in patients who received high-dose intravenous medication [2]. Tooth extraction, maxillofacial surgery, and invasive periodontal surgery are trigger factors for MRONJ in individuals using these medications, while local infection remains one of the most important risk factors in the development of MRONJ [3]. According to the most recent consensus by the American Association of Oral and Maxillofacial Surgeons (2022), for a diagnosis of MRONJ to occur there must be three elements [4]: (1) Current or previous treatment with antiresorptive therapy alone or in combination with immune modulators or antiangiogenic agents; (2) Exposed bone or bone that can be probed through an intraoral or extraoral fistula(e) in the maxillofacial region that has persisted for more than eight weeks; (3) No history of radiation therapy to the jaws or metastatic disease to the jaw.

The Australian Oral and Dental Therapeutic Guidelines [5] describe the criteria used to assess the risk of developing MRONJ before a bone-invasive dental procedure in patients treated with an antiresorptive or antiangiogenic drug (Figure 1). Patients are classified as being at high or low risk of developing MRONJ based on the duration of exposure, indication for treatment, and additional risk factors.

The typical presentation of MRONJ is unhealed mucosa and exposed necrotic bone in a previous surgical site. Other symptoms of MRONJ may include jaw or tooth pain, swelling, and neuropathy. It can subsequently lead to difficulty in chewing, infection, and poorer dental treatment outcomes. The pathophysiology of MRONJ is not completely understood; however, the literature suggests it may involve suppression of bone remodelling, inflammation, altered immune function, inhibition of angiogenesis, and soft tissue toxicity [4].

Drugs implicated in the pathogenesis of MRONJ are antiresorptive agents, antiangiogenic agents, and the mammalian target of rapamycin (mTOR) inhibitors [1]. BPs and the receptor activator of nuclear factor kappa-Β ligand (RANKL) inhibitors are antiresorptive drugs. BPs mediate their antiresorptive effects by inhibiting osteoclasts, thereby limiting bone resorption and maintaining bone density [6]. There are two main groups of BPs: nitrogen-containing (N-BP) and non-nitrogen (NN-BP) containing, with the former being more potent and widely used [6]. Denosumab (DEN) is a monoclonal antibody that limits bone resorption by inhibiting RANKL, a key mediator of osteoclast function [7].

Antiangiogenic drugs such as bevacizumab (BEV) and sunitinib (SUN) block the formation of new blood vessels by reducing the action of vascular endothelial growth factor (VEGF) and tyrosine kinases [8]. VEGF also plays a role in the regulation of osteoclast cells, and inhibition of this factor may provide insight into the mechanisms contributing to an increased risk of MRONJ in these patients [9]. mTOR is a key regulator in the growth, proliferation, and metabolism of cells [10].

Despite the debilitating nature of MRONJ, the mechanism by which it occurs is not well understood. Hence, this systematic review aims to summarise the existing literature on the in vitro effects exerted by antiresorptive and antiangiogenic drugs on cells isolated from oral bone tissue, oral mucosa, and periodontal ligament from the oral cavity. 

### Objectives

The specific questions addressed in this systematic review are as follows: What are the in vitro effects of different antiresorptive and antiangiogenic drugs on the survival, proliferation, and migration of oral mucosal and bone cells?Do the in vitro effects of antiresorptive and antiangiogenic medications on oral mucosal and bone cells change in a dose-dependent manner?Are there potential therapeutic agents that may alleviate the in vitro effects of antiresorptive and antiangiogenic drugs on oral mucosal and bone cells?

## 2. Materials and Methods

This systematic review uses the Preferred Reporting Items for Systematic Reviews and Meta-Analyses (PRISMA) guidelines as a framework for the reporting of results [11].

### 2.1. Study Selection

#### 2.1.1. Inclusion Criteria

Studies were included if they exposed cells from the oral mucosa, associated bone, or periodontal tissues to antiresorptive or antiangiogenic drugs, in vitro. Only articles written in English were considered and there was no restriction placed on the date of publication.

#### 2.1.2. Exclusion Criteria

Studies were excluded if they were review articles, commentaries, conference abstracts, opinion articles, letters to the editor, case series, case reports, or retracted.

#### 2.1.3. Screening Process

A search of the literature was completed, followed by screening based on the inclusion criteria above. The process undertaken is depicted in Figure 2.

Step 1: A literature search was completed on the 24th of May 2022 in Medline (Ovid), Embase (Ovid), Scopus, and Web of Science by three independent reviewers (AH, HO, and RG). The search strategy and syntax used can be found in Appendix B. The results were imported into Covidence (Veritas Health Innovation, Melbourne, Australia) where duplicates were automatically removed, and further screening was conducted.

Step 2: Papers were screened by title and abstract by two independent reviewers (RH and RG). An initial pilot screening of 40 articles was completed before screening the remaining 351 papers. Any discrepancies between the two reviewers were resolved in consultation with the research supervisor (AC).

Step 3: The remaining papers were split into two groups of 78 articles each. Each group of articles was assessed for eligibility by two independent reviewers (JH, XH, HO, and TD). Data were extracted from articles that met the inclusion criteria and were tabulated. Discrepancies were resolved in the same way as described in Step 2.

### 2.2. Statistical Analysis

In order to assess inter-rater agreement during screening, Cohen’s kappa was calculated using IBM Statistics 27 (SPSS). The absolute percentage agreement between raters was also calculated. 

### 2.3. Risk of Bias

The 59 studies included in this systematic review were analysed using an adapted Office of Health Assessment and Translation (OHAT) risk of bias tool for internal validity [12]. The OHAT risk of bias tool utilises 10 questions and an additional “other potential threats to internal validity” category to assess potential bias in human and non-human animal studies. For the studies in this systematic review, it was agreed that eight questions and other biases “statistical analysis” and “adherence to study protocol” were relevant. For each of these questions, the risk of bias was assigned as either “definitely low”, “probably low”, “probably high”, or “definitely high”. The questions assessed can be found in Appendix C.

## 3. Results

### 3.1. Data Selection and Collection

Out of the 850 articles retrieved, 459 duplicates were automatically removed by Covidence. This left 391 articles for screening. Cohen’s kappa statistic and the absolute percentage agreement for the initial pilot title and abstract screening was 0.75 (95% confidence interval [CI]: 0.54–0.96) and 87.50%, demonstrating a good level of agreement. Cohen’s kappa statistic and absolute percentage agreement for the complete title and abstract screening was 0.87 (95% CI: 0.82–0.92) and 93.61%, indicating a very good level of agreement. Cohen’s kappa scores and absolute percentage agreement for the two groups involved in screening the full texts were 0.46 (95% CI: 0.26–0.66) and 74.36% and 0.77 (95% CI: 0.63–0.91) and 88.46%, respectively. This demonstrates a moderate and good level of inter-rater agreement. Fifty-nine articles [13,14,15,16,17,18,19,20,21,22,23,24,25,26,27,28,29,30,31,32,33,34,35,36,37,38,39,40,41,42,43,44,45,46,47,48,49,50,51,52,53,54,55,56,57,58,59,60,61,62,63,64,65,66,67,68,69,70,71] were included in this systematic review, based on the inclusion criteria (Appendix D). 

### 3.2. Quality Assessment (OHAT results)

A bias rating of “definitely low” was found in domains “selection”, “performance”, “detection”, “selective reporting” and “other biases” (93.22%, 57.63%, 45.76%, 35.59%, and 22.03%, respectively) (refer to Figure 3). The risk of bias was found to be “probably low” in the domains “selection”, “performance”, “attrition/exclusion”, “detection”, “selective reporting” and “other biases” (6.78%, 40.68%, 100.00%, 52.54%, 50.85%, and 71.19%, respectively). A bias rating of “probably high” was found in the domains “performance”, “detection”, “selective reporting”, and “other biases” (1.69%, 1.69%, 11.86%, and 5.08%, respectively). A bias rating of “definitely high” was found in the domains of “selective reporting” and “other biases” (1.69% in both domains). The full OHAT risk of bias assessment can be found in Appendix A.

### 3.3. Study Characteristics

All 59 selected articles investigated the in vitro effects of antiresorptive or antiangiogenic drugs on cells that were derived from the periodontal ligament, oral mucosa, or associated bone. The articles were published between 2008 and 2022 and originated from a variety of different countries. The full data extraction table can be found in Appendix A.

BPs were, by far, the most commonly used drug class (56 articles, 95%). Table 1 provides further characterisation of the different drugs used in the included studies.

A majority of the 59 articles included studied cells derived from soft tissue only (48 articles, 81%), followed by cells derived from bone tissue only (nine articles), and soft tissue and bone (two articles). Most of the included studies used primary cells (37 articles, 63%), followed by cell lines (21 articles, 36%) and primary cells and cell lines (one article). In most cases, cells were derived from humans (54 articles, 92%), whereas in a minority (five studies) they originated from animals. Table 2 shows the different cell types utilised and the frequency of their use in the 59 included papers. A wide variety of different assays were used to study the effects of the drugs on the cells (refer to Appendix A).

### 3.4. Use of Bisphosphonates on Cells Derived from Soft Tissue

Several trends were observed when cells were exposed to N-BPs (refer to Table 3). These trends included an increased rate of apoptosis (25 articles) and a decrease in proliferation (17 articles), migration (23 articles), metabolism (11 articles), and viability (38 articles). Six articles reported changes in cell morphology following exposure to N-BPs [19,24,26,34,69,71]. Three articles reported an increase in inflammation and the expression of inflammatory markers chemokine ligand 2 (CCL2) and interleukin (IL)-6 in gingival fibroblasts (GFs), following exposure to N-BPs [20,58,60]. In contrast, Yuan et al. [68] found that there was a decrease in IL-6 production following exposure to N-BPs, whereas Tamai et al. [57] found that N-BP exposure did not alter cytokine production. Ten studies included clodronate (CA), an NN-BP. Six studies found that CA did not alter at least one of migration, apoptosis, proliferation, or viability [29,39,45,46,63,71]. In contrast, four other studies found that all of the cellular responses tested were affected [32,41,62,68]. The concentration of CA required to decrease cell viability, proliferation, and migration was higher than the dose of N-BPs required to achieve the same outcome. 

### 3.5. Use of Nitrogen-Containing Bisphosphonates on Cells Derived from Bone Tissue

Several studies have demonstrated that N-BPs exert similar effects on cells derived from bone tissue (refer to Table 4). A decrease in cell proliferation (five articles), migration (one article) and viability (three articles), and an increased rate of apoptosis (three articles) were observed [17,25,28,40,70]. However, unlike soft tissue, there was no study that investigated the effects of N-BPs on cell metabolism. 

### 3.6. Use of Cytoprotectants/Rescue Drugs on Cells Treated with Nitrogen-Containing Bisphosphonates

Eight studies included the use of geranylgeraniol (GGOH), an isoprenoid, to determine if it could reverse the effects of the N-BPs (refer to Table 5) [22,26,31,34,46,69,70,71]. Seven of these studies concluded that GGOH increased viability and migration of N-BP treated cells [22,26,34,46,69,70,71] and three of the studies showed increased proliferation [22,31,70]. One study concluded that GGOH reduced the rate of apoptosis and increased bone nodule formation in alveolar osteoblasts (AOBs) treated with N-BP [70]. Three articles also found that GGOH recovered cell morphology and the actin cytoskeleton to an appearance that resembled non-treated control cells [26,34,71]. One study suggested ozone gas plasma may help to reduce the genotoxic effects of N-BPs and promote wound healing in N-BP-treated cells [16]. Melatonin increased proliferation in zoledronate (ZA) treated cells, whilst another study found that hydroxyapatite (HA) maintained metabolic activity in cells treated with ZA and pamidronate (PA) [21,50]. One study using a cryoprotectant dexrazoxane found that it increased cell metabolism and restored cell morphology [24]. Walter et al. [63] found that low-level laser treatment (LLLT) increased the viability of oral keratinocytes (OKs) and GFs treated with N-BPs.

### 3.7. Use of Other Antiresorptive and Antiangiogenic Drugs

Hoffman et al. [27] found that BEV and SUN altered the expression of genes involved in osteogenesis in human AOBs. There was decreased expression of ALPL and SPARC in both groups, whilst cells treated with SUN also exhibited a decrease in COL1A1 expression. It was also observed that exposure to both drugs resulted in an increase in matrix metallopeptidase 1 (MMP1) and secreted phosphoprotein 1 (SPP1). Angiogenic marker expression was also altered with a decrease in platelet-derived growth factor β polypeptide (PDGFB) expression in cells treated with BEV and an increase in VEGFR2 expression in cells treated with SUN. Hoffman et al. [27] also reported that cells treated with SUN had increased expression of the proinflammatory cytokines IL-1β, IL-8, and TNF-α. One article found that a mouse monoclonal antibody to RANK-L did not affect the viability of gingival fibroblasts or alter the expression of the pro-apoptotic genes assessed (Bak, Bad, Bax, Bim) [36]. Another study investigated the effects of DEN on the proliferation of dental follicle cells, finding that the drug had no effect [42]. Yuan et al. [68] observed that DEN did significantly reduce cell migration following 48 h of treatment. They also observed that, like other antiresorptives tested, there was an increase in VEGF and IL-1β expression following treatment with DEN.

## 4. Discussion

### 4.1. Overview

Antiresorptive and antiangiogenic drugs are amongst the most commonly prescribed medications for the management of osteometabolic diseases and cancer [72]. However, they have been implicated in the development of MRONJ, first described by Marx in 2003 with PA and ZA [73]. The list of drugs reported to be associated with MRONJ has grown to include other major drug groups such as RANKL-inhibitors, antiangiogenics, and mTOR inhibitors [74]. This has led to research aiming to elucidate the nature and pathophysiology of MRONJ, specifically in its unique localisation to the maxillofacial region. This systematic review aims to provide a summary of the existing in vitro literature and translate findings into informing guidelines for clinical practice. 

### 4.2. Drugs Investigated

N-BPs were overwhelmingly the most commonly investigated drugs, in particular, ZA, which was investigated in 46 of the 59 studies. ZA has the greatest potency and duration of action amongst BPs in clinical use and is a low-cost generic drug [72]. It is the first-line therapy for the prevention and management of osteoporosis and is frequently used as an adjunct therapy in cancer patients. It is also used extensively in patients who have contraindications for oral BPs such as alendronate (AA) or risedronate (RA) [72]. This may explain the extensive research behind ZA compared to other BPs. DEN and BEV were among the least investigated drugs, appearing in only four of the 59 studies, possibly due to the high cost of obtaining monoclonal antibodies for research.

### 4.3. Comparing N-BPs and NN-BPs

N-BPs have a greater antiresorptive potency compared to NN-BPs [75]. This disparity in potency is illustrated in a number of studies that show a general consensus that N-BPs have a greater influence on cell viability, apoptosis, and cell migration at low concentrations [29,45,46,62,63,71]. Pabst et al. [45] found that ZA decreased cell viability starting at 5 μM; however, CA showed no significant influences on cell viability at this concentration. CA only significantly decreased viability at 50 μM. Similar results were also found in studies by Walter et al. [62] and Pabst et al. [46]. These observations may be explained by the absence of nitrogen atoms in their R2 side chain structures [76]. The R2 side chain determines the antiresorptive power of the drugs and influences the mechanism of action [76]. NN-BPs, such as CA and etidronate, antagonise the cellular energy pathways through the liberation of methylene, a toxic NN-BP metabolite, which facilitates the formation of methylene ATP analogues [77]. These analogues compete with ATP, therefore, compromising ATP-utilising processes [77]. In comparison, N-BPs interfere with the mevalonate pathway (MVP) by inhibiting farnesyl pyrophosphate synthase (FPPS). This reduces the prenylation of proteins and inhibits the activation of small GTPases, thereby altering cell signalling and impairing normal cell function which results in reduced proliferation, migration, and alterations to cell morphology [34]. In these pathways, NN-BPs rapidly undergo metabolism; however, due to the presence of nitrogen, N-BPs are not readily metabolised and as a result, are more likely to accumulate in the tissues [76,78]. This increases its duration of action and therefore, its potency. 

Differences in potency also exist between different N-BPs with ZA being reported to be the most potent followed by AA and PA [14]. This is due to the different three-dimensional R2 side chain structures [76]. Unlike most R2 side chains which are straight chains, ZA consists of a nitrogen-containing heterocyclic structure, which gives rise to its greater potency. 

### 4.4. Pathways Implicated in Bisphosphonate-Related Cellular Cytotoxicity

Numerous studies found that antiresorptive and antiangiogenic drugs are genotoxic and cytotoxic to oral tissues, impairing cell proliferation, metabolism, viability, and migration, altering cell morphology and inducing apoptosis and inflammation in a dose- and time-dependent manner [15,25,41,42,44,48,51,52,59,70].

Transforming growth factor beta (TGF-β) was found to induce the differentiation of fibroblasts into myofibroblasts, thereby promoting wound healing, cell migration, viability, and proliferation [33]. However, Ziebart et al. [71] found that ZA suppresses TGF-β activity, thereby impairing the re-epithelization of the tissues in the oral mucosa as well as reducing wound healing. N-BPs were found to elevate reactive oxygen species (ROS) production, which is thought to occur due to the inhibition of farnesyl pyrophosphate (FPP). N-BP-induced ROS production likely contributes to decreased cell migration and proliferation through the regulation of cell growth factors and signalling pathways [59]. Cells treated with ZA were found to be retained in the G0/G1 phase of cell division [48]. Wang et al. [65] showed that the gene expression of Cyclin D1 was downregulated in orofacial mesenchymal stem cells treated with PA. One possible explanation for decreased cell proliferation and retention in G0/G1 is the downregulation of Cyclin D1 gene expression, as it regulates the progression of cells from the G1 to the S phase of the cell cycle [79].

Scheper et al. [51] showed that oral tissues treated with ZA had elevated expression of genes involved in the intrinsic (TNF, TRAF, death domain) and extrinsic (BCL, IAP, Caspase) apoptotic pathways, potentially explaining the increased rate of apoptosis observed in multiple studies. Increased caspase-3 and 9 activity, in cells treated with ZA and PA, as well as increased ROS production in N-BP-treated cells likely induced apoptosis in these cells [40,51,65,66].

An increase in inflammatory protein and gene expression was found in multiple studies. Basso [20] showed an increased synthesis of chemokine ligand 2 (CCL2) involved in the recruitment and polarisation of macrophages during inflammation. Yuan et al. [68] found an increased expression of proinflammatory cytokines IL-6, IL-8, and TNF in cells treated with AA in the presence of lipopolysaccharide (LPS). 

Treatment with N-BPs showed changes in cell morphology that are consistent with cell stress, including enlarged nuclei and an altered structure [24]. N-BPs were found to alter the actin cytoskeleton, possibly explaining changes in morphology [71].

### 4.5. Pathways Implicated in Anti-RANKL mAb-Related Cellular Cytotoxicity

Kuroshima et al. [36] found that mouse monoclonal antibodies to RANKL (anti-RANKL mAb), which have been shown to be comparable to DEN in humans, significantly suppressed osteoclast numbers. It was found that osteoclastogenesis was not affected but rather the pro-apoptotic factors Bad, Bax, and Bim were significantly upregulated. Therefore, the authors theorised that the decrease in osteoclast numbers was due to apoptosis. Furthermore, it was found that anti-RANKL mAb did not affect GFs, indicating that DEN may primarily affect the bone tissues and not the soft tissues in MRONJ development. Mosch et al. [42] suggested that DEN causes toxicity in a receptor-mediated manner. The authors found that DEN had a toxic effect on mesenchymal stem cells at 10 μM and 20 μM; however, the vitality of dental follicle cells, which do not have the RANK receptor, was not affected. 

### 4.6. Pathways Implicated in Antiangiogenic-Related Cellular Cytotoxicity

BEV and SUN were both found to downregulate (COL1A1, ALPL, SPARC) genes involved in bone mineralization [27]. Conversely, both drugs were found to elevate genes involved in bone degradation (MMP-1); therefore, they may potentially prevent new bone formation and healing, thereby contributing to the development of MRONJ [27]. Additionally, PDGFB can induce osteoblast differentiation and therefore its downregulation by BEV may further reduce the bone-forming capacity in MRONJ [27]. SUN was found to increase the production of a variety of different inflammatory cytokines which the authors speculate may interfere with bone healing and repair [27].

### 4.7. Potential Adjunct Therapeutics

N-BPs inhibit the farnesyl pyrophosphate of the MV [80]. In osteoclasts, this interferes with geranylgeranylation and results in the inactivation of the MVP and reduced bone turnover [80]. Geranylgeraniol (GGOH) is a downstream metabolite that can replenish geranylgeranyl diphosphate and prevent the inhibition of osteoclast formation and bone resorption [80]. Kim et al. [31] found that PA-induced senescence in oral keratinocytes (OKs) in an MVP-dependent manner via geranylgeranylation, suggesting that this premature senescence of oral mucosal cells may cause defective soft-tissue wound healing, contributing to the development of MRONJ in patients. Included papers studying the effects of GGOH found that it ameliorated the in vitro effects of BPs [22,26,31,32,46,69,70,71]. In vivo studies have also shown the positive effects of GGOH on bisphosphonate-treated bone cells. Nagaoka et al. [81] found that although injections of ZA and LPS decreased the bone mineral density and volume in mice, simultaneous injections of GGOH alongside ZA and LPS significantly increased the bone mineral density and volume. Similarly, another in vivo study in rats has shown the positive effects of GGOH on oral soft tissue cells, finding decreased inflammation and infection [82]. Based on the current in vitro and in vivo evidence, GGOH could be a promising adjunctive therapy for N-BPs.

Most of the other isoprenoids tested showed little efficacy when compared to GGOH. Hagelauer et al. [26] found that only eugenol had an effect on ZA-treated cells, and increased wound healing capacity; however, it had a significantly lower positive influence compared to GGOH. The ineffectiveness of farnesol (FOH) suggested that the lack of protein geranylgeranylation, rather than protein farnesylation, leads to N-BP-induced suppression of cell functions as little FOH goes through the geranylgeranylation pathway [26]. This is supported by Kim et al. [31] who found PA and FOH-treated OKs have a similar decrease in proliferation when compared to PA-treated OKs. However, PA and GGOH-treated OKs gained back approximately half of their proliferative capacity [31]. 

### 4.8. Effects on Non-Orally Derived Cells

Soft tissue cells derived from the rest of the body, including non-oral epithelial cells and human umbilical vein endothelial cells (HUVECs), had similar responses to orally derived cells. Exposure to N-BP and NN-BPs significantly reduced cell proliferation and migration and increased apoptosis [26]. However, some studies found that orally derived soft tissue cells exhibited more pronounced effects in response to N-BPs, displaying slower wound healing time, reduced growth, and significantly higher ROS production [58,59].

MRONJ is uniquely localised to maxillofacial bones. Maxillofacial bones develop exclusively from the neuroectoderm and are primarily formed by intramembranous ossification, whereas peripheral bones develop from the mesoderm and are formed by both endochondral and intramembranous ossification [83,84,85]. Although the structure of these different bones is identical, functional differences in bone turnover and the mechanical properties at different anatomic sites may exist. Because jaw-bone marrow stem cells (BMSCs) differ from peripheral bone BMSCs in development and phenotype, it is hypothesised that BPs modulate cell function and the osteogenic potential of BMSCs in a skeletal site-specific pattern [25].

Gong et al. [25] found that ZA inhibited the proliferation of both jaw BMSCs and iliac and tibial BMSCs in a dose-dependent manner; however, for the latter two, proliferation increased at low ZA concentrations before decreasing dose-dependently. When compared to the untreated group, the ZA-treated jaw BMSC complex showed less scaffold degradation, collagen fibres, and bone-like tissue formation, whereas the ZA-treated iliac BMSC complex showed more new bone-like tissue formation [25]. These findings are indicative of ZA having a greater influence on jaw BMSCs, compared to iliac and tibial BMSCs, resulting in inhibition of bone formation.

### 4.9. The ‘Drug Holiday’

In a 2022 position paper, the American Association of Oral and Maxillofacial Surgeons discussed the controversial nature of the ‘drug holiday’ that has been suggested by many researchers [4]. They reported the results of a systematic review looking into the efficacy of drug holidays in clinical studies, which found no consensus on the timing and duration of the drug holiday [86]. Some studies suggested that the drug holiday should be dependent on individual patient factors, whereas others reported that a drug holiday had no effect at all. The inadequacy of the cessation of BP treatment before invasive surgery may be due to the high binding affinity of BPs to HA in vitro, and long functional half-lives in vivo [87]. None of the articles included in this systematic review supported the use of a drug holiday. One in vivo study using animal models found that the frequency and severity of MRONJ were reduced after a 6-week preoperative and 8-week postoperative drug holiday [88]. However, this was achieved in combination with antibiotic prophylaxis, smoothening of sharp edges of bone, and wound closure with mucoperiosteal flaps. This evidence suggests that there is potential for drug holidays in combination with other standard post-operative procedures to be beneficial. Further in vivo and in vitro studies are required to understand the contribution of these drugs to the development of MRONJ and subsequently produce a guideline that can be implemented by clinicians. 

### 4.10. Limitations in the Existing Literature

In conducting this systematic review, several limitations were identified. None of the studies investigated whether there would be compounding effects if cells were treated with both antiresorptive and antiangiogenic drugs. For example, this would be clinically relevant for cancer patients who are prescribed both drug types. The concentration of drugs used in these in vitro studies likely does not reflect the maximum concentration found in oral tissues at clinically administered doses, making it difficult to determine the extent of cytotoxicity at therapeutic doses. Although some studies reported potential pathways to explain the observed cytotoxic effects, the exact mechanisms as to how these drugs cause MRONJ to develop are largely unknown. All of the studies also only investigated the short-term effects of these drugs on cells. However, these drugs are normally prescribed for long periods in patients, therefore highlighting the need for more long-term in vitro and in vivo studies where the concept of a drug holiday could also be investigated. Several studies researching the rescue potential of various compounds found promising results for potential co-treatments to reduce the risk of MRONJ and warrant further investigation. Finally, although BPs were researched extensively, there were few studies on monoclonal antibodies and no studies on mTOR inhibitors. This highlights the importance of future research into these drug classes, potential adjunct therapies, and only after that, the need to progress to in vivo studies.

## 5. Conclusions

This systematic review found that antiresorptive and antiangiogenic drugs consistently displayed cytotoxic effects in vitro in a dose and time-dependent manner. The current body of literature primarily features the inhibition of cell proliferation, viability, migration, and the induction of apoptosis and cell cycle arrest in vitro. A majority of the current literature focuses on N-BPs, highlighting the importance of future research into the growing list of drug classes implicated in the development of MRONJ. Evidence for compounds that are able to alleviate the observed cytotoxic effects highlight a promising avenue for future research as co-therapies for the prevention and treatment of MRONJ. There remains limited literature surrounding the concept of a drug holiday, highlighting a need for further animal and human studies.

## Figures and Tables

**Figure 1 biomolecules-13-00973-f001:**
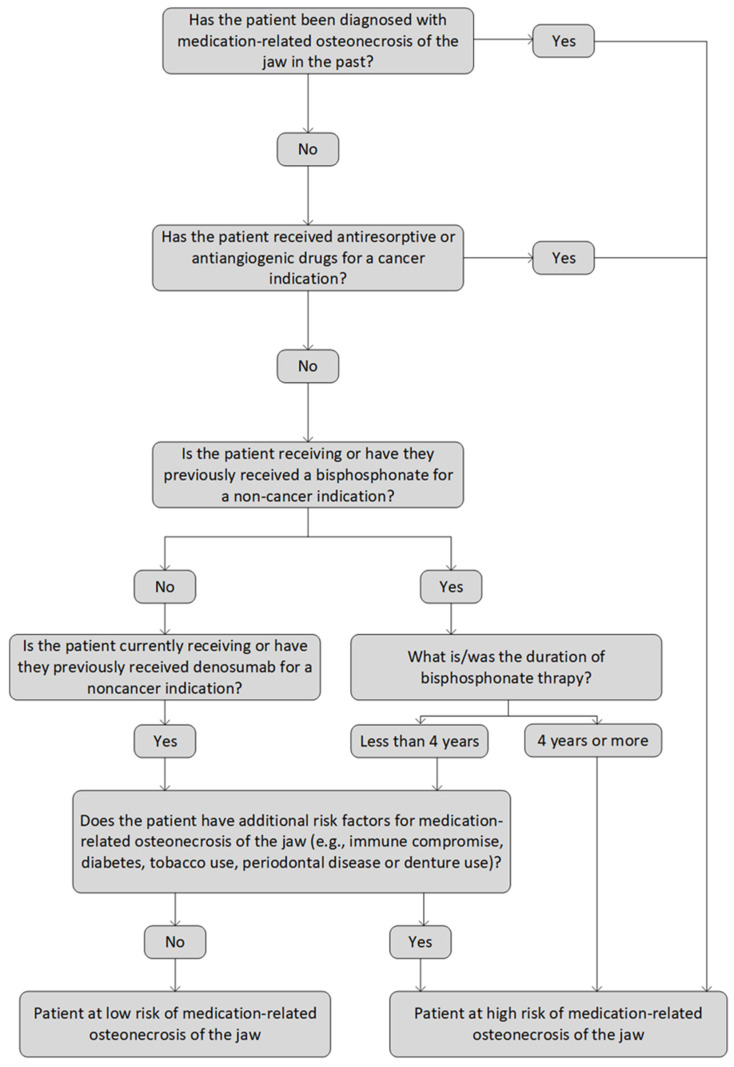
MRONJ risk assessment based on the Australian Oral and Dental Therapeutic Guidelines.

**Figure 2 biomolecules-13-00973-f002:**
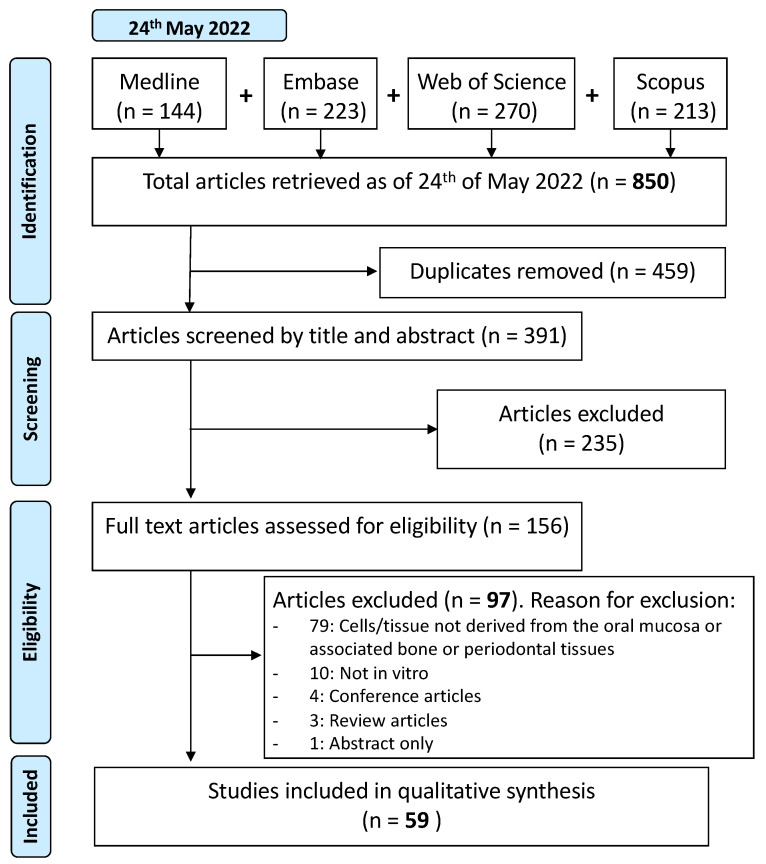
Process undertaken to select studies for the systematic review into the effect of antiangiogenic and antiresorptive drugs on oral-cavity-derived cells.

**Figure 3 biomolecules-13-00973-f003:**
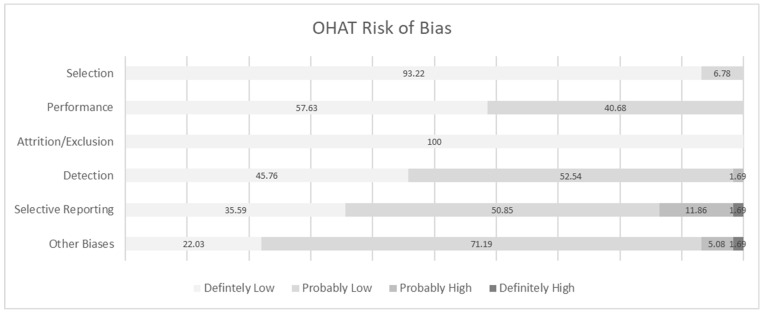
Summary of the risk of bias assessment of the 59 included articles based on amended OHAT guidelines.

**Table 1 biomolecules-13-00973-t001:** Frequency of use of different antiresorptive and antiangiogenic drugs in the included articles.

Class of Antiresorptive or Antiangiogenic Drug	Type of BP	Frequency of Use
BPs	ZA	46 (78%)
	PA	17 (29%)
	AA	14 (24%)
	IA	7 (12%)
	RA	2 (3%)
	CA	10 (17%)
BEV	N/A	1 (2%)
DEN	N/A	2 (3%)
SUN	N/A	1 (2%
Mouse monoclonal antibody to RANKL	N/A	1 (2%)

Abbreviations: BPs, bisphosphonates; BEV, bevacizumab; DEN, denosumab; SUN, sunitinib; RANKL, receptor activator of nuclear factor kappa-B ligand; ZA, zoledronate; PA, pamidronate; AA, alendronate; IA, ibandronate; RA, risedronate; CA, clodronate; N/A, not applicable.

**Table 2 biomolecules-13-00973-t002:** Frequency of use of soft tissues and bone tissues in the included articles.

Tissue Source	Cell Types	Frequency of Use
Soft tissue	GFs	32 (54%)
	OKs	10 (17%)
	PDLFs	5 (8%)
	PDLSCs	4 (7%)
	OFs	2 (3%)
	SCC-9	1 (2%)
	SCC-15	1 (2%)
	GECs	1 (2%)
	OFMSCs	1 (2%)
Bone tissue	AOBs	6 (10%)
	BMCs	4 (7%)
	JPO	1 (2%)

Abbreviations: GFs, gingival fibroblasts; OK, oral keratinocytes; PDLFs, periodontal ligament fibroblasts; PDLSCs, periodontal ligament stem cells; OFs, oral fibroblasts; SCC-9, squamous cell carcinoma-9; SCC-15, squamous cell carcinoma-15; GECs, gingival epithelial cells; OFMSCs, orofacial mesenchymal stem cells; AOBs, alveolar osteoblasts; BMCs, bone marrow cells; JPO, jaw periosteum.

**Table 3 biomolecules-13-00973-t003:** Cellular responses to different orally derived soft tissues when exposed to BPs.

Exposure Characteristics			Cellular Response				
Type of BP	Drug	Cell type	Apoptosis	Proliferation	Migration	Metabolism	Viability
N-BP	ZA	GFs	↑ 8	↓ 7	↓ 5	↓ 5	↓ 13 =1
		OKs	↑ 3	↓ 2	↓ 3	↓ 1	↓ 6
		PDLFs	↑ 2	N/A	↓ 2	↓ 1	↓ 1
		PDLSCs	↑ 2	↓ 1	↓ 1	N/A	N/A
	PA	GFs	↑ 1	↓ 2	↓ 1	↓ 2	↓ 4
		OKs	↑ 2	↓ 2	↓ 3	↓ 1	↓ 4
		OFs	↑ 1	N/A	N/A	N/A	↓ 1
	AA	PDLFs	↑ 3	N/A	↓ 2	N/A	↓ 1
		PDLSCs	N/A	N/A	↓ 1	N/A	N/A
		OKs	N/A	↓ 1	N/A	N/A	↓ 1
		GFs	N/A	↓ 2	N/A	N/A	↓ 1
	IA	OKs	↑ 2	N/A	↓ 2	N/A	↓ 2
		PDLFs	↑ 1	N/A	↓ 1	N/A	↓ 1
		GF	N/A	N/A	↓ 1	N/A	↓ 3
	RA	PDLSCs	N/A	N/A	↓ 1	N/A	N/A
NN-BP	CA	GFs	=1	N/A	=1	N/A	↓ 2 =2
		OKs	↑ 1 =1	↓ 1 =1	↓ 1 =1	N/A	↑ 1 ↓ 2 =1

Values indicate the number of studies these results were found in. Abbreviations: BPs, bisphosphonates; N-BPs, nitrogen-containing bisphosphonates; NN-BPs, non-nitrogen-containing bisphosphonates; ZA, zoledronate; PA, pamidronate; AA, alendronate; IA, ibandronate; RA, risedronate; CA, clodronate; GFs, gingival fibroblasts; OKs, oral keratinocytes; PDLFs, periodontal ligament fibroblasts; PDLSCs, periodontal ligament stem cells; OFs, oral fibroblasts; N/A, not applicable; ↑, increase; ↓, decrease; =, no change.

**Table 4 biomolecules-13-00973-t004:** Cellular responses to different orally derived bone tissues when exposed to N-BPs.

Exposure Characteristics		Cellular Response			
Drug	Cell Type	Apoptosis	Proliferation	Migration	Viability
ZA	AOBs	↑ 3	↓ 2	↓ 1	↓ 2
	BMCs	N/A	↓ 2	N/A	N/A
	JPO	N/A	=1	N/A	N/A
PA	AOBs	N/A	↓ 1	N/A	↓ 1

Values indicate the number of studies these results were found in. Abbreviations: ZA, zoledronate; PA, pamidronate; AOBs, alveolar osteoblasts; BMCs, bone marrow cells; JPO, jaw periosteum. N/A, not applicable; ↑, increase; ↓, decrease; =, no change.

**Table 5 biomolecules-13-00973-t005:** Main findings of adjunct treatments on different orally derived cell types and the frequency of use in the included articles.

Adjunct Treatment	Frequency of Use	Cell Types	Main Findings
GGOH	8 (14%)	GFs	↑ migration, proliferation, adhesion, migration, and wound healing capacity. Rescued cell morphology.
		OKs OFs	Partially recovers proliferation.
		AOBs	↑ cell viability, proliferation, and migration. ↓ rate of apoptosis. Bone nodule formation. Rescued cell morphology.
		GECs	↑ cell viability and improves cell morphology.
FOH	4 (7%)	GFs	=wound healing capacity, viability, or morphology.
		OKs OFs	=cell proliferation.
		AOBs	Partial restoration of cell viability
MOH	1 (2%)	GFs	=wound healing capacity, viability, or morphology.
EU	1 (2%)	GFs	↑ wound healing capacity, = viability, or morphology.
SQ	1 (2%)	GFs	=wound healing capacity, viability, or morphology.
R-(+) limonene	1 (2%)	GFs	=wound healing capacity, viability, or morphology.
Ozone gas plasma	1 (2%)	GFs	↓ genotoxic effect and ↑ wound healing.
PRGF	1 (2%)	GFs AOBs	↑ cell proliferation, ↓apoptosis, and inflammation.
HA	1 (2%)	GFs OKs	↑ metabolic rate.
Dexrazoxane	1 (2%)	GFs	↑ metabolic rate and rescued cell morphology.
rhPDGF-BB	1 (2%)	GFs	Partially rescue cell migration, proliferation, and adhesion.
TGFβ-1	1 (2%)	GFs	=cell viability.
OME	1 (2%)	GFs	=cell viability.
BCP	1 (2%)	GFs	↑ metabolic rate and cell migration.
Melatonin	1 (2%)	PDLSCs	↑ cell proliferation.
NAC	1 (2%)	PDLFs	↓ ROS production.
LLLT	1 (2%)	GFs	↑ viability only seen in cells treated with IA, no effect observed on cells treated with CA, PA, and ZA.
		OKs	↑ viability in cells treated with N-BPs, no effect on cells treated with NN-BPs.
EGF	1 (2%)	OKs	↑ viability in cells treated with low concentrations of ZA, no effect seen with higher doses. Increased cell migration.
Arachidonic acid	1 (2%)	PDLFs	↑ ROS production in ZA-treated cells.
Lipid A	1 (2%)	GFs	In combination with AA-induced ↑ IL-6 and IL-8.
PRP and PRF	1 (2%)	GFs	↑ migration and viability of ZA-treated cells.

Abbreviations: GGOH, geranylgeraniol; FOH, farnesol; MOH, menthol; EU, eugenol; SQ, squalene; PRGF, plasma rich in growth factors; HA, hydroxyapatite; rrPDGF-bb, recombinant human-platelet-derived growth factor BB; TGFB-1, transforming growth factor beta 1; OME, omeprazole; BCP, biphasic calcium phosphate; NAC, N-acetyl-cysteine; LLLT, low-level laser treatment; EGF, epidermal growth factor; PRP, platelet rich plasma; PRF, platelet-rich fibrin; GFs, gingival fibroblasts; OKs, oral keratinocytes; OFs, oral fibroblasts; AOBs, alveolar osteoblasts; GECs, gingival epithelial cells; PDLSCs, periodontal ligament stem cells; PDLFs, periodontal ligament fibroblasts; ROS, reactive oxygen species; IA, ibandronate; CA, clodronate; PA, pamidronate; ZA, zoledronate; N-BPs, nitrogen-containing bisphosphonates; NN-BPs, non-nitrogen-containing bisphosphonates; AA, alendronate; IL-6, interleukin 6; IL-8, interleukin 8; ↑, increase; ↓, decrease; =, no change.

## Data Availability

The data that support the findings of this study are available from the corresponding author upon reasonable request.

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
