# Peer review of "In Vitro Cytotoxicity of Antiresorptive and Antiangiogenic Compounds on Oral Tissues Contributing to MRONJ: Systematic Review"

_biomolecules, 2023, doi:10.3390/biom13060973_

Round 1
Reviewer 1 Report
The review is interesting and well conducted. However, there are some points that need to be corrected:
1. In the introduction: the incidence of MRONJ was lower than what we know so far from the literature "between 1 in 10,000 and < 1 in 100,000 patient-treatment years". Please revise that refer to a more updated reference.
2. Also in the introduction "Tooth extraction, maxillofacial surgery, and invasive periodontal surgery are trigger factors for MRONJ in individuals using these medications". Local infection is one of the most important risk factors of MRONJ, this factor has to be mentioned as many extraction cases with MRONJ are likely to be triggered by local infection which indicated extraction and not the extraction itself. Please refere to Otto et al. review about MRONJ and infection.
3.the causes of exclusion have to be mentioned in Prisma chart.
4. please add percentage when possible beside numbers in your results
Author Response
The review is interesting and well-conducted. However, there are some points that need to be corrected:
Comment 1: In the introduction: the incidence of MRONJ was lower than what we know so far from the literature "between 1 in 10,000 and < 1 in 100,000 patient-treatment years". Please revise that refer to a more updated reference.
Reply: Thank you for your valuable comment. We updated the reported incidence values according to a very recent study by Schwech et al, published in Feb 2023. The respective sentence now reads: “The overall incidence of MRONJ varies widely across different studies, being reported from 0.01% after oral use of low-dose bisphosphonates to 14.4% in patients who received high-dose intravenous medication”
Comment 2: Also in the introduction "Tooth extraction, maxillofacial surgery, and invasive periodontal surgery are trigger factors for MRONJ in individuals using these medications". Local infection is one of the most important risk factors of MRONJ, this factor has to be mentioned as many extraction cases with MRONJ are likely to be triggered by local infection which indicated extraction and not the extraction itself. Please refere to Otto et al. review about MRONJ and infection.
Reply: We thank this reviewer for this comment. We added local infection as per your suggestion and now the section reads: “Tooth extraction, maxillofacial surgery, and invasive periodontal surgery are trigger factors for MRONJ in individuals using these medications, while local infection remains one of the most important risk factors in the development of MRONJ.”
Comment 3: the causes of exclusion have to be mentioned in Prisma chart.
Reply: We thank this reviewer for this comment. The PRISMA chart has been completely re-done and the causes of exclusion have now been added as per your request.
Comment 4: please add percentage when possible beside numbers in your results
Reply: We thank this reviewer for this comment. We added the percentage equivalent of each value reported, with the exception of table 3 and 4, where the addition would have created overwhelming tables.
Reviewer 2 Report
This is a very nice review with clear and concise language. Minor changes may be considered:
1. Although abbreviations are already introduced, it may be easier for the reader to re-introduce the full name of some less known compounds, such GGOH, in the discussion section.
2. Even though bisphosphonates and denosumab were found to have similar effects in soft tissue and bone derived cells, authors should make a point that studies from bone cells are likely more clinically relevant given our in vivo knowledge that these drugs primarily target bone and particularly osteoclasts. This underlines the need for adjunct therapeutic molecules to be examined mostly in bone cell lines in future studies.
Author Response
This is a very nice review with clear and concise language. Minor changes may be considered:
Comment 1: Although abbreviations are already introduced, it may be easier for the reader to re-introduce the full name of some less known compounds, such GGOH, in the discussion section.
Reply: Thank you for your valuable comment. Thank you for this valuable comment. We re-introduced the fully spelled out term “geranylgeraniol” at the beginning of the discussion as per your request (Page 14).
Comment 2: Even though bisphosphonates and denosumab were found to have similar effects in soft tissue and bone derived cells, authors should make a point that studies from bone cells are likely more clinically relevant given our in vivo knowledge that these drugs primarily target bone and particularly osteoclasts. This underlines the need for adjunct therapeutic molecules to be examined mostly in bone cell lines in future studies.
Reply: Thank you for your valuable comment. We agree with this reviewer that these two points should have been also highlighted in the take home message from our manuscript. Therefore we included these points in the conclusion section of the manuscript, which now reads: “ This systematic review found that antiresorptive and antiangiogenic drugs consistently displayed cytotoxic effects in vitro in a dose and time dependent manner. The current body of literature primarily features the inhibition of cell proliferation, viability, mi-gration, and the induction of apoptosis and cell cycle arrest in vitro. A majority of the current literature focuses on N-BPs, highlighting the importance of future research into the growing list of drug classes implicated in the development of MRONJ. Evidence for compounds that are able to alleviate the observed cytotoxic effects highlight a promising avenue for future research as co-therapies for prevention and treatment of MRONJ. Multiple studies have found comparable results in both soft tissue and bone tissue cell lines. However, since bisphosphonates have a high affinity for bone tissue, those studies which used bone tissue are likely to be more clinically relevant. Therefore, the authors feel that more studies investigating the effect of adjunct therapeutics should be conducted using bone tissue. There remains limited literature surrounding the concept of a drug holiday, high-lighting a need for further animal and human studies”.